# A Clean and Health-Care-Focused Way to Reduce Indoor Airborne Bacteria in Calf House with Long-Wave Ultraviolet

**DOI:** 10.3390/microorganisms12071472

**Published:** 2024-07-19

**Authors:** Luyu Ding, Qing Zhang, Chaoyuan Wang, Chunxia Yao, Feifei Shan, Qifeng Li

**Affiliations:** 1Information Technology Research Center, Beijing Academy of Agriculture and Forestry Sciences, Beijing 100097, China; dingly@nercita.org.cn (L.D.);; 2National Engineering Research Center for Information Technology in Agriculture (NERCITA), Beijing 100097, China; 3National Innovation Center of Digital Technology in Animal Husbandry, Beijing 100097, China; 4Department of Agricultural Structure and Bioenvironmental Engineering, College of Water Resources and Civil Engineering, China Agricultural University, Beijing 100083, China; 5Intelligent Equipment Research Center, Beijing Academy of Agriculture and Forestry Sciences, Beijing 100097, China

**Keywords:** closed calf house, emission rate, size distribution, microbial composition, health improvement

## Abstract

Long-term exposure to a relatively high concentration of airborne bacteria emitted from intensive livestock houses could potentially threaten the health and welfare of animals and workers. There is a dual effect of air sterilization and promotion of vitamin D synthesis for the specific bands of ultraviolet light. This study investigated the potential use of A-band ultraviolet (UVA) tubes as a clean and safe way of reducing airborne bacteria and improving calf health. The composition and emission characteristics of airborne bacteria were investigated and used to determine the correct operating regime of UVA tubes in calf houses. Intermittent exceedances of indoor airborne bacteria were observed in closed calf houses. The measured emission intensity of airborne bacteria was 1.13 ± 0.09 × 10^7^ CFU h^−1^ per calf. *Proteobacteria* were the dominant microbial species in the air inside and outside calf houses. After UVA radiation, the indoor culturable airborne bacteria decreased in all particle size ranges of the Anderson sampler, and it showed the highest reduction rate in the size range of 3.3–4.7 μm. The results of this study would enrich the knowledge of the source characteristics of the airborne bacteria in intensive livestock farming and contribute to the environmental control of cattle in intensive livestock production.

## 1. Introduction

Intensive animal farming is widespread throughout the world currently and has become a major emission source of air pollutants into the atmosphere, including odor, ammonia, greenhouse gases, and airborne bacteria [1,2,3]. Airborne bacteria and their byproducts, such as endotoxins, could attach to solid or liquid particles and be aerosolized to be bioaerosols, which consist of living organisms and suspend in the air [4,5]. Emissions of airborne bacteria or bioaerosol from livestock houses affect not only the health of animals, but also the quality of their products [6,7]. The results from previous surveys showed a positive relationship between calf respiratory disease and airborne bacteria in calf barns in the winter [8]. Besides causing diseases, air microorganisms can also contaminate freshly milked milk through post-secretory contamination, which will ultimately cause severe consequences from the perspective of public health [9]. Furthermore, long-term exposure to a relatively high concentration of airborne bacteria or bioaerosols in the workplace or residential place would result in a number of potential threats to respiratory health, including infectious diseases, allergies, and so on [10,11,12,13,14].

Due to potential threats of airborne bacteria or bioaerosols to animals and human, environment sterilization is necessary in intensive farms to reduce the microorganisms in the air inside animal houses, especially for newborn animals like calf. Disinfectants or detergent sprays are the prevailing approaches adopted in livestock houses in the presence of animals [15]. Nevertheless, prolonged usage of disinfectants or detergents would lead to microbial resistance and potential adverse effects on the environment [16,17]. In addition, spray sterilization will increase the air humidity. Dairy animals tolerate moisture poorly, while humidity in closed animal houses is normally very high and even sometimes tends to become saturated in winter [7]. Due to the poor resistance of calves to the external environment, they are easily attacked by pathogenic microbes, and high air humidity would increase the incidence probability of calves. Therefore, it is necessary to find a new method to reduce indoor airborne bacteria without increasing the air humidity in the house.

Ultraviolet (UV) is separated into four bands, which are UVA (400–320 nm), UVB (320–380 nm), UVC (280–200 nm), and vacuum ultraviolet (VUV, 200–80 nm) [18], which have different biological effects with different wavelength bands. Regularly, C-band ultraviolet (UVC) rays are usually used in ultraviolet sterilization lamps. However, UVC cannot be used in animal houses, especially in the presence of animals, as it has a strong biological effect and may cause DNA damage to animals [19,20]. UVA accounts for about 95% of the UV radiation that reaches the Earth’s surface. It has been shown that UVA has a certain bactericidal effect and is not as biologically destructive as UVC [21]. The proper use of certain doses and wavelength bands of UV even has some health-care effects, such as promoting the formation of vitamins, and treating refractory immune-related dermatitis [22,23]. Hence, UVA is considered a novel sterilization method to improve the safety of animals, plants, and humans. For example, Wang et al. [24] developed a sterilization device equipped with ultraviolet (UVA)-light-emitting diodes to disinfect the nutrient solution residual liquid of a hydroponic system in a greenhouse. The results showed a sterilization rate of 94% at a wavelength close to 300 nm, a working power of 30 w, and a sterilization time of 70 s. Qi et al. [25] use the combination of UVA-LED (365 nm) and low-concentration peroxymonosulfate for wastewater sterilization from recirculating aquaculture. They proved the potential strategy to introduce the mercury-free UV-LED sterilization process in aquaculture. Based on these studies, UVA irradiation may be a potential clean and safe way for air sterilization in the presence of animals in calf houses.

There is a source specificity for airborne bacteria as microbial populations originating from diverse emission sources are often heterogeneous and represent a mixture of local and long-range transported biological particles [26,27]. In the perspective of a better understanding of bacteria bioburden, the characterization of the microbial components and their size distribution may facilitate the understanding of the different emission sources and the bioaerosol dispersal patterns present in complex workplaces such as livestock facilities [28,29]. This would be helpful for the proper use of indoor sterilization to prevent livestock pathogen disease and assess the threats of intensive farming to public health security [2,30,31]. Although intensive farming is a very important emission source, limited research can be found to clarify the characteristics, flux, and changes in distribution of airborne bacteria or bioaerosols [32]. In China, there is normally a solid floor in the housing system for dairy and beef cattle. The excrement of cattle remains on the floor and is not cleaned up in time. Coupled with the trampling of animals in daily activities, this increases the probability of gasification of the viable biological contaminants from excrement or the animals themselves to be airborne and become airborne bacteria or bioaerosols. It is important to investigate the characteristics of airborne bacteria or bioaerosol from cattle houses to determine the proper operation regime of UVA indoor sterilization and to further assess its health risk.

Hence, this study investigated the characteristics of airborne bacteria in a closed calf house and examined the potential use of UVA in indoor air sterilization in the presence of calves. Specifically, microbial components and their size distribution of airborne bacteria were characterized in the calf house. The spatial and temporal variation in airborne bacteria, as well as their emission rate, was examined to determine the correct operating regime of UVA tubes in calf houses. The effect of UVA irradiation was evaluated through both the reduction in airborne bacteria and the behavior of calves. The results of this study would enrich the knowledge and understanding of the distribution patterns and source characteristics of the airborne bacteria or bacterial bioaerosol from the emission source of intensive livestock farming and contribute to the clean production of cattle or calves in enclosed housing systems.

## 2. Materials and Methods

### 2.1. Experimental Calf House

The present study was carried out in an enclosed calf house located in the suburban area of Shijiazhuang in Hebei Province. The calf house is 37.0 m long and 17.5 m wide, with a double-pitch roof and tunnel ventilation in cold seasons. There are four frequency conversion exhaust fans. All fans were operated, and the airflow rate was manually controlled during the experiment. There are a total of 10 pens in the calf house, symmetrically distributed in two rows with a middle aisle (Figure 1). The two pens near the door are 7.0 m long, and the remaining eight pens are 6.0 m long, both with a width of 5.0 m. There is a lying area (laid with bedding material) and a walkway area (solid concrete floor) in the pen. Each pen keeps 7 to 8 calves, ranging in age from 2 to 4 months. The calves are fed three times a day at 6:00, 14:00, and 20:00, respectively. There is solid concrete floor in the calf house and the bedding area is laid with rice straw. Bedding materials are replaced daily in the morning, and the walkway area in the pen is flushed with water to remove manure twice a day (once in the morning and once in the afternoon). The calf house is disinfected with spray once a day every morning during this study as regular. Thus, microbial sources of airborne bacteria mainly include the animal itself, excreta of animals, outdoor air brought into the house through ventilation, and indoor facilities in the experimental house such as feed, bedding material, etc.

### 2.2. Description of Tests and Sampling Strategies

Three tests were conducted to investigate the characteristics of airborne bacteria from calf houses and the bactericidal effect of UVA irradiation. These three tests were the spatial variation test, the temporal variation test, and the bactericidal effect test. Indoor and outdoor air temperature (T), relative humidity (RH), and the ventilation rate in the calf house were recorded throughout the whole experimental duration. Indoor and outdoor T and RH were logged by a recorder (Apresys 179A-TH, Los Angeles, CA, USA) every 10 min at a height of 2 m at three points. The running frequency of exhaust fans was kept stable during the whole day and was manually recorded to obtain the daily ventilation rate by the correlation between the running frequency and airflow rate of each fan [33]. The outdoor airborne bacteria were sampled at a location near the air inlet of the calf house at a height of 1.2 m, with two replicates. The sampling strategies of indoor airborne bacteria were slightly different between three different tests due to different goals, which are described in detail in the following sections.

#### 2.2.1. Spatial Variation Test

The spatial variation test was carried out mainly to examine the spatial distribution, microbial composition, and size character of airborne bacteria in the calf house. In the spatial variation test, there were one outdoor point and seven indoor points to sample airborne bacteria (Figure 1). The six-stage Anderson samplers (TISCH, TEI Corp., Hamilton, OH, USA) with disposable nutrient agar plates (every 1000 mL of nutrient agar contains: peptone 10.0 g, beef extract 3.0–5.0 g, sodium chloride 5.0 g, agar powder 12.0–14.0 g) were used to sample airborne bacteria. Sampling was conducted over five consecutive days from 11:00 to 13:00 at a height of 1.2 m with a flow rate of 28.3 L min^−1^. Each point was sampled for about 2 min and had two replicates.

Samples were stored in an insulated box (0–10 °C) after collection and transported to the laboratory for incubation within 1 h. After incubation (37 °C, 24 h), colonies in sampling plates were counted with a counter (Icount-30F, Xunshu Technology Co., Hangzhou, China). Correction of colony numbers and concentration calculation of airborne bacteria were performed using Equations (1) and (2), respectively.
(1)$$Pr=N\times (\frac{1}{N}+\frac{1}{N-1}+\ldots +\frac{1}{N-r+1})$$
where *Pr* is the number of colonies after correction, CFU m^−3^; *N* is the number of sampling holes at each level of the six-stage Anderson sampler; and *r* is the actual number of colonies, CFU.
(2)$$C=\frac{{Pr}_{1}+{Pr}_{2}+{Pr}_{3}+{Pr}_{4}+{Pr}_{5}+{Pr}_{6}}{t\times F}\times 1000$$
where *C* is the concentration of airborne bacteria, CFU m^−3^; *Pr_x_* is the number of colonies after calibration of each sampler stage, *x* = 1, 2, 3, 4, 5, 6; *t* is the duration of sampling, min; and *F* is the sampling flow rate of the sampler, 28.3 L min^−1^.

There is a specificity of microbial components from different emitting sources of airborne bacteria [28]. To identify the microbial composition of airborne bacteria in calf houses, indoor and outdoor air (points 7 and W), as well as manure, were sampled by the Coriolis^®^ µ sampler (Bertin Technologies, Montigny-le-bretonneux, France) at a height of 1.2 m (medium: 15 mL ultrapure water; duration: 10 min; flow rate: 300 L min^−1^).

The collected samples were sent for 16S rRNA full-length sequencing analysis. The total genomic DNA from the samples was extracted using the CTAB/SDS method, and the DNA’s concentration and purity were assessed by DNA agarose gel electrophoresis. Then, a series of procedures were conducted on the DNA samples, including 16S full-length amplification, the construction of the SMRT Bell library, and sequencing on the PacBio platform. Combined with bioinformation analysis, the samples were classified by barcode for clustering and species classification analysis to study environmental microbial diversity and community composition differences.

#### 2.2.2. Temporal Variation Test

The temporal variation test was carried out mainly to examine the emission rate, diurnal variation in concentration, and emission for airborne bacteria in the calf house and to analyze the proper irradiation time of UVA in the following test. Combining the spatial distribution of the concentration of airborne bacteria, one outdoor point and two indoor points were used to sample airborne bacteria (Figure 1). Sampling was conducted at 8:00, 9:00, 11:00, 13:00, 15:00, and 17:00 on five consecutive days using the same sampling and incubation methods as those in the spatial variation test. The emission rate of airborne bacteria from the calf house was calculated based on the ventilation rate and concentration difference between indoors and outdoors using Equation (3).
(3)$$F=(C_{in}-C_{out})\times V$$
where *F* is the emission rate of airborne bacteria from the calf house, CFU h^−1^; *C_in_* and *C_out_* are the indoor and outdoor concentrations of airborne bacteria, respectively, CFU m^−3^; and *V* is the ventilation rate of the calf house, m^3^ h^−1^.

#### 2.2.3. Bactericidal Effect Test of UVA

Selecting two pens at the same concentration level as the treatment and control based on the spatial variation test (Figure 1), an experimental test was conducted to investigate the bactericidal effect of ultraviolet irradiation of the A-band in the presence of calf. The schematic diagram of the treatment is shown in Figure 2. Eight UVA lighting tubes (340 nm, 1200 mm length, 40 W) were installed at the top of the pen (2.3 m above ground level) with an interval of 1 m. Based on the results of the temporal variation test, UVA irradiation was performed 2 h per day during 8:00–10:00 in the treatment pen.

After the one-week treatment, airborne bacteria in the treated pen and in the pen of control were sampled from 8:00 to 12:00 every half an hour over seven consecutive days using the sampling and incubation method as before. As there is a natural decay of airborne bacteria, the dynamic inactivation rate (DIR) of airborne bacteria was defined and calculated using Equations (4) and (5) to quantify the bactericidal effect of UVA.
(4)$$\eta_{1}=\frac{{c_{0-}c}_{n}}{c_{0}}\times 100\%$$
(5)$$DIR=\frac{c_{0}^{\prime }\left(1-\eta_{1}\right)-c_{n}^{\prime }}{c_{0}^{\prime }\left(1-\eta_{1}\right)}\times 100\%$$
where η_1_ is the natural decay rate of airborne bacteria, %; *c*_0_ and *c_n_* are the concentration of airborne bacteria initial and at certain moment in the control pen, respectively, CFU m^−3^; *DIR* is the dynamic inactivation rate of airborne bacteria for UVA, %; and *c*_0_′ and *c_n_*′ are the concentration of airborne bacteria initial and at certain moment in the treatment pen, respectively, CFU m^−3^.

The health and welfare of animals are largely reflected in their behaviors. Calves perform abnormal repetitive behaviors when they undergo feeding restrictions, barren housing environments, or compromised animal welfare [34]. In addition, grooming is a positive social behavior common in calves in relatively good welfare conditions. To further evaluate the improvement in UVA irradiation on calf health and welfare, the occurrence of grooming and abnormal behavior of calves was consecutively recorded during 9:00–10:00 in the control and treatment pens. The record of abnormal behaviors (except grooming) of calves is shown in Table 1, and their occurrence was calculated by the ratio between the number of calves with aberrant behavior cases and the total number of calves in the pen.

### 2.3. Data Processing and Analysis Methods

The significance test of difference was performed to analyze the spatial variation in the culturable airborne bacteria inside the calf house and compare the concentration of culturable airborne bacteria as well as the observed behaviors of calves between the control and UVA treatments. The average concentration and emission rate of the measured airborne bacteria were expressed as mean ± standard error in this study. To investigate the microbial diversity of the collected samples, cluster analysis of the operational taxonomic unit (OTU) was performed based on the results of 16S rRNA full-length sequencing.

## 3. Results and Discussion

### 3.1. Spatial and Temporal Characteristics of Airborne Bacteria in Calf House

The measured indoor T and RH were 13.01 ± 2.46 °C and 45.02 ± 15.78%, and the outdoor T and RH were 11.47 ± 2.98 °C and 41.04 ± 18.72% during the spatial and temporal variation tests. The measured ventilation rate was 35,079 ± 8118 m^3^ h^−1^ and 22,543 ± 6834 m^3^ h^−1^ during the spatial and temporal variation tests, respectively.

#### 3.1.1. Concentration and Emission Rate

Airborne bacteria were sampled as shown in Figure 1. As shown in Figure 3, the sampled culturable bacteria ranged from 2.43 × 10^3^ to 2.01 × 10^5^ CFU m^−3^ in the enclosed calf house. According to the industrial standards of agriculture in China [36], the proposed guideline for indoor bacteria is below 2 × 10^4^ m^−3^ for cattle. The average culturable bacteria was 1.49 ± 0.91 × 10^4^ CFU m^−3^ in different sampling locations, below the proposed guideline. However, due to the uneven distribution of airborne bacteria and insufficient ventilation, there is a spatial variation in different sampling locations, and the culturable bacteria exceeded the proposed guidelines in some regions (Figure 3a). These regions tend to be relatively far away from the exhaust fans (e.g., sampling location 5) or on the feed-spreading aisle (e.g., sampling location 7). Theoretically, there should be a lower concentration of culturable bacteria in sampling locations 5 and 6 as they are near the air inlet of the calf house. However, relatively more culturable airborne bacteria were measured in these locations in this study. This may be because the air leaks from the calf house near the exhaust fans, resulting in the poor tunnel effect of negative pressure. A temporal variation was also observed for the sampled culturable bacteria and averaged 5.72 ± 2.28 × 10^4^ CFU m^−3^ at different sampling times, which highly exceeded the proposed guideline (Figure 3b). This may be due to the insufficient ventilation and disturbance of daily events (e.g., manure cleaning).

Furthermore, it can be seen that the culturable bacteria sampled in the days examining the temporal variation (Figure 3b) were much higher than those sampled in the days examining the spatial variation (Figure 3a). This is because the ventilation rate was about 37% larger in the days examining the spatial variation, and airborne bacteria sampling was conducted during 11:00–13:00, which had the lowest culturable bacteria according to the temporal variation in Figure 3b. There was a peak observed in the morning and one in the afternoon, respectively, for the culturable bacteria sampled at different times of the day. The time of these two peaks coincides with the time of daily activity for workers (e.g., feeding, pen cleaning). The higher peak was observed in the morning. This would be due to the replacement of bedding material. Bedding material is replaced once a day in the morning.

Matković et al. [37] quantified the airborne bacteria in a free-stall dairy barn and demonstrated that the mean total bacterial count inside the dairy house ranged from 8.81 × 10^4^ CFU m^−3^ to 1.26 × 10^5^ CFU m^−3^ from morning to evening. Similarly, the lowest bacteria count was found at noon in the daily variation. Gladding et al. [3] investigated the concentration and composition of airborne bacteria from intensive pig and poultry farms. The results showed that culturable bacteria outdoor were 1.77 × 10^2^ to 1.09 × 10^4^ CFU m^−3^, 2.65 × 10^2^ to 1.56 × 10^4^ CFU m^−3^, and 1.24 × 10^2^ to 4.33 × 10^3^ CFU m^−3^ when air temperature was 10.6–24.2 °C in the layer farm, broiler farm, and swine farm, respectively. The measured outdoor culturable bacteria ranged from 6.81 × 10^2^ to 9.09 × 10^3^ CFU m^−3^ in this study, close to those measured in swine farms but lower than those in poultry farms. The concentration or count of culturable bacteria is highly dependent on the airflow rate or ventilation condition of the surroundings. As shown by Xiang et al. [38], the concentration of airborne bacteria rose from 2 × 10^3^ CFU m^−3^ to 1 × 10^4^ CFU m^−3^ along the increased length of incoming air passing away in the long axis of manure belt poultry houses. Thus, it is more important to know the emission rate to compare or assess the emission intensity of airborne bacteria.

In this study, the emission rate of airborne bacteria was estimated based on the airflow rate in the calve house and the concentration difference between indoor and outdoor culturable bacteria. The estimated emission rate from the investigated calf house was averaged at 7.89 ± 0.61 × 10^8^ CFU h^−1^, which is equivalent to the emission intensity of 1.13 ± 0.09 × 10^7^ CFU h^−1^ per calf. Figure 4 shows the daily variation in emission rates from the calf house. A similar variational tendency was found between the emission rate and the counts of culturable bacteria. The emission rate during 11:00–13:00 was very stable, while increasing at first and then decreasing during 8:00–11:00 and 13:00–17:00. There is very limited information on the emission rates of airborne bacteria from animal buildings, while concentrations of airborne microorganisms and dust in livestock production systems are highly correlated. Resuspension is an important mechanism for the transmission of airborne bacteria. As particle matter would be the carrier of microorganisms, in–out air motion, worker activities and any factor that may cause animals to be more active would increase the amounts of solid and liquid particles, and, in turn, increase the amounts of airborne bacteria [1,39]. This may be part of the explanation of the observed diurnal variation.

#### 3.1.2. Size Distribution

The composition and hazard of airborne bacteria to animals or humans varied with different carrier sizes. The larger the carrier size, the faster sedimentation, and the shorter the traveling distance or existence time in the air [40,41]. In this study, the proportion of airborne bacteria in different carrier sizes was obtained according to the culturable bacteria retained in different stages of the six-stage Anderson sampler. Figure 5 shows the size distribution of airborne bacteria measured in the calf house. The top three highest proportions of carriers were distributed in the sizes of >7.0 µm, 2.1–3.3 µm, and 3.3–4.7, with a proportion of 23.32%, 22.42%, and 19.12%, respectively. The lowest proportion was distributed in the size range of 0.65–1.1 µm, which accounts for 4.87%.

Small aerosol particles can exist in the air for a long time and can eventually be inhaled by people and animals along the air flow, causing the spread of disease. Reports demonstrate that airborne bacteria with a carrier size of less than 5.0 μm can easily enter the deep respiratory tract, and those with a carrier size of less than 2.5 μm can enter the bronchioles, alveolar sacs, and alveoli of the lung [42]. Hu et al. [42] investigated the size distributions of bioaerosols in an egg production facility and found that 60–80% of in-house bacteria were in the respirable size range (≤4.7 μm). Similarly, in this study, 76.69% of the measured airborne bacteria had a carrier size of less than 5.0 μm, which indicates that most of the airborne bacteria emitted from calf houses can be inhaled into the respiratory tract and pose a threat to the health of animals and humans.

Comparing with the existing literature, the results of this study are consistent with the existing research showing that airborne bacteria emitted from animal houses are mainly distributed on coarse particles whose particle size is over 2.1 μm [42,43,44]. Li et al. [43] investigated the spatiotemporal variations in the association between particulate matter and airborne bacteria in concentrated layer feeding operations and found that particles with diameters ranging from 2.1 to 4.7 µm carried the most airborne bacteria. Zhong and Wang [45] compared the size distribution of airborne bacteria in different animal houses to assess their potential health hazard risk. The results showed that the highest proportion of airborne bacteria was measured in the first stage of the six-stage Anderson sampler. The proportion of inhalable particles that carry bacteria (particle size < 8.2 µm) accounted for 67%, 76%, and 74% for poultry houses, pig houses, and cattle houses, respectively. The proportion of the second peak was in the range of 3.3–4.7 µm, and the distribution proportion in the range of 0.65–1.1 µm was the smallest.

#### 3.1.3. Microbial Composition and Diversity

Based on the 16Sr RNA sequencing, the top 10 phyla with high abundance are listed in Table 2 for the microbial composition of the samples from indoor air, outdoor air, and the fresh manure of calves. *Firmicutes* (abundance > 10%) were the dominant bacteria in the air inside and outside the calf house and in the fresh manure of calves. *Proteobacteria* were the dominant microbial species in the air inside and outside calf houses. This may be related to the fact that the airborne bacteria in calf houses are mainly affected by the activities of the calf and daily operations. The main source of these is animal excrement.

Samples in the air of the calf house contained 1.51% *Staphylococcus*, and *Staphylococcus aureus* in the genus was one of the most important pathogens causing cow mastitis. In the studies by Popescu et al. [46] and Matković et al. [37], *Staphylococcus* was also detected in dairy houses. In addition, there are 2.34% *Aerococcus* and 0.90% *Pseudomonas* measured in the air of the calf house. *Aerococcus viridis*, a representative strain of *Aerococcus*, can cause endocarditis, urinary tract infection, septic arthritis, meningitis, and other diseases. *Pseudomonas* is distributed in soil, water, and various planting objects, and some are pathogenic to humans and animals, such as *Pseudomonas aeruginosa*, *Pseudomonas putidis*, *Pseudomonas fluorescens*, and so on. Hence, it may contain pathogenic bacteria or selective pathogenic bacteria in the air of the calf house.

The microbial diversity of indoor and outdoor air, as well as that of fresh manure, was analyzed according to 16Sr RNA sequencing. The sequences were clustered into operational taxonomic units (OTUs) with 97% agreement of gene sequence taxa, resulting in 2655 OTUs. As shown in Figure 6, there are 981 OTUs clustered in indoor air in total. Among which, there are 417 OTUs common in indoor and outdoor air, 389 OTUs common in indoor air and fresh manure, and 203 OTUs common in all three kinds of samples. This suggests that there are other important sources for the microorganisms in indoor air besides manure and outdoor air. About one-third of the microbial species are unique to indoor air.

The saturation curves of the species number were analyzed for the three different kinds of samples, which increased with the sequencing depth and tended to flatten eventually, indicating that the current sequencing depth has essentially covered all species in the sample and was sufficient for the analysis of microbial diversity. To evaluate the similarity between samples from the three different sources, unweighted group mean clustering analysis (UPGMA) was performed using the abundance information of OTUs and the weighted distance matrix. As shown in Figure 7, it can be seen that the samples from indoor air and outdoor air were clustered together, indicating a high similarity of microbial species between them.

### 3.2. Bactericidal Effect of UVA Irradiation

Ultraviolet radiation attenuates as the distance from the lamp increases. As shown in Figure 8b, the relationship between ultraviolet radiation intensity and vertical distance was investigated before the bactericidal effect test. The distance was about 1.20 m from the UV tubes to the height near the back and muzzle of calves (standing height), where ultraviolet irradiation attenuates to about 2.64 µW cm^−2^. Laboratory simulation tests were conducted to examine the impacts of the radiation intensity and duration of UVA on the decay of airborne bacteria (generated by an aerosol dispenser with the medium of the extracting solution from cow manure and deionized water) before this study. Based on the results of laboratory simulation tests (unpublished data), there was no significant difference in the decay of airborne bacteria (about 40%) between the radiation intensities of 2, 500, and 1000 µW cm^−2^ when the irradiation duration was long enough (60 min) and below 1000 µW cm^−2^.

#### 3.2.1. Reduction in Indoor Airborne Bacteria

The measured concentration of airborne bacteria in treatment and control pens is shown in Figure 9. The concentration in the control reflected the background concentration of airborne bacteria in the calf house, and the concentration in the treatment reflected the bactericidal effect of UVA irradiation. Before UVA irradiation and two hours after stopping UVA exposure, the concentrations of airborne bacteria in the pens between the treatment and control were very close to each other (at 8:00 and 12:00 in Figure 9). During UVA irradiation, the concentration of airborne bacteria in the treatment pen was significantly lower than that in the control pen, showing a good bactericidal effect on airborne bacteria in the measured concentration range of around 2.95 × 10^4^ CFU m^−3^.

As mentioned in Section 2.2.3, the DIR of airborne bacteria was calculated to evaluate the bactericidal effect of UVA. The average DIR for airborne bacteria was 32.13% during the 2 h irradiation. It showed a sustaining bactericidal effect of UVA irradiation within two hours after stopping irradiation, and the average DIR was very close to each other in the range of 8:00–10:00 (during UVA irradiation), 10:00–12:00 (2 h after stopping irradiation) and 8:00–12:00 (whole examination period). The average DIR for airborne bacteria was 32.60% within 2 h of stopping irradiation, and the overall average DIR during 8:00–12:00 was 32.40%. This may be caused by the production of reactive oxygen species such as superoxide anion radicals (O_2_·) and ozone under ultraviolet irradiation [47]. The DIR of UVA irradiation seems to be related to the concentration of airborne bacteria. It tended to have a higher DIR when there was a relatively higher concentration of airborne bacteria. The top three highest DIRs were observed when the top three highest concentrations were measured in the control pen. The DIR reached its highest value (43.15%) when the UVA irradiation was carried out for 30 min during 8:00–10:00, in which the highest concentration of airborne bacteria appeared according to the diurnal variation in Figure 3b. After the end of lighting, the efficiency of continuous action decreased with the increase in time, and the inhibition effect of UVA irradiation on bioaerosol was gradually weakened. Combined with the diurnal variation in airborne bacteria in the calf house in Section 3.1.1, it is recommended to operate UVA irradiation under a radiation intensity of about 2 µW cm^−2^ in the two peak concentration periods for 2 h, i.e., 8:00–10:00 and 14:00–16:00.

The bactericidal effect was compared on airborne bacteria of different carrier sizes according to the number of cultured colonies retained on different stages of the Anderson sampler after 2 h of UVA irradiation in the control and treatment pens. As shown in Figure 10, the number of cultured colonies in the treatment pen was significantly lower than those in the control in different carrier size ranges after 2 h of UVA irradiation. It showed the highest reduction rate was in the size range of 3.3–4.7 μm and the lowest reduction rate was in the size range of 0.65–1.1 μm, which were 64.47% and 22.73%, respectively. There was a similar reduction rate in the size ranges of >7.0 μm and 3.3–4.7 μm, which were 43.90% and 41.22%, respectively. It tends to have a better bactericidal effect on coarse carriers. A higher reduction rate was observed in the size range of over 3.3 μm than in the range of 1.1–3.3 μm. This is partly consistent with the previous result that the bactericidal effect of UVA irradiation is related to the concentration of airborne bacteria. As shown in Figure 10, 57.36% of the sampled airborne bacteria were distributed in a size range of over 3.3 μm. Among them, UVA showed a better bactericidal effect on the size ranges of 4.7–7 μm and 3.3–4.7 μm, whose reduction rates were both over 50%. Li et al. [48] used ultraviolet lamps with an intensity of 70 µW cm^−2^ and 110 µW cm^−2^ for 8 h of air irradiation to examine their bacteria reduction effect in a rural wastewater treatment station. The results showed that the bacteria with a carrier size range of 1.1–4.7 µm were reduced by 54.2% and 61.5%, respectively. Similar results were observed, although different wave bands, intensities, and exposure durations of ultraviolet were used between this study and that of Li et al. [48].

#### 3.2.2. Health Improvement for Enclose Housed Calves

The tests of the significance of the difference were performed on the recorded behaviors of calves in the control and treatment pens. Table 3 shows the results of the tests before and after UVA irradiation. Grooming helps ease the tension between calves and form or reinforce social bonds between individuals. Calves often perform abnormal oral behaviors, including tongue rolling and NNOM, when opportunities to perform feeding behaviors are restricted. There are no significant differences in grooming and NNOM behaviors between the control and treatment before and after UVA irradiation. This indicates that the operation of UVA radiation when disinfecting calves in the house will not affect or alter the expression of social behavior or normal feeding behavior. Calves showed a headbanging behavior because of a runny nose. Headbanging and cough generally occur when there is contaminated air or due to the invasion of pathogenic microorganisms. Before UVA irradiation, there was no significant difference in cough behavior between the control and treatment, and headbanging was observed in the treatment pen but not in the control pen. After UVA irradiation, there was no headbanging observed in the treatment, but headbanging appeared in the control pen instead. Furthermore, a relatively lower cough was shown in the trend in the treatment pen. This suggests that the proper irradiation of calves with UVA can improve their respiratory health due to the bactericidal effect of UVA. The irradiation of UVA may cause skin fiber aging and melanin deposition [49]. After UVA irradiation, a relatively higher trend in itching behavior in the treatment pen was observed. Hence, further systematic studies are recommended and needed to conduct a systematic assessment of the impacts of UVA irradiation on the health and performance of calves. It would be important to evaluate the safety and benefit of UVA irradiation in the presence of calves to perform air sterilization before technology extension.

## 4. Conclusions and Future Work

This study explored the potential use of UVA irradiation as a clean and relatively safe way for air sterilization in the presence of animals in the enclosed calf house. The spatial and temporal variation tests were conducted to characterize the concentration and carrier size distribution, microbial composition, and diversity, as well as the emission rates of culturable airborne bacteria inside the calf house in winter and to seek the proper operation regime of UVA irradiation in the calf house. Afterwards, the bactericidal effect test was conducted to examine the reduction rate of indoor airborne bacteria from UVA irradiation. The impact of UVA irradiation on the behavioral expression of calves was examined as an indicator to assess the influence of UVA on animal health and welfare. The measured culturable airborne bacteria ranged from 2.43 × 10^3^ to 2.01 × 10^5^ CFU m^−3^, exceeding the Chinese standard limit of air quality requirement inside the calf house in some time even though spray sterilization was performed once a day every morning. Uneven spatial distribution and diurnal variation in culturable airborne bacteria were observed inside the calf house, and two peak concentration periods appeared at 9:00 and 15:00, in which the concentration of airborne bacteria normally exceeded the standard requirement. The operation regime of UVA sterilization was recommended as twice a day and two hours for each during 8:00–10:00 and 14:00–16:00 in the enclosed calf house. When the radiation intensity of UVA (340 nm, working power of 40 W) was about 2.64 µW cm^−2^ at the standing height of near the back and muzzle of calves, the average reduction rate was 32.13% during two hours of irradiation at an indoor concentration of around 2.95 × 10^4^ CFU m^−3^. The reduction rate showed a tendency to be positively related to the concentration of indoor culturable airborne bacteria, and there is a sustaining bactericidal effect of UVA irradiation within two hours of stopping irradiation. In this study, the calves treated with UVA sterilization showed a tendency for a lower occurrence of cough and a higher occurrence of itching. A long-term survey was suggested to further evaluate the safety and benefit of UVA irradiation in the presence of calves before the extension of UVA sterilization.

Due to the deficiencies of the present study, more systematic assessments need to be conducted to evaluate the extension of the use of UVA irradiation. In future work, further research should be carried out in three aspects. Firstly, an in-depth analysis of microbial components before and after irradiation should be conducted. This should consider not only the microbial composition and diversity tests on indoor air but also the influence on the normal and beneficial microbiota of animals. The present study analyzed the phylum of airborne bacteria. Genus and species are suggested to be further examined to analyze their sensitivities to UVA. Secondly, a long-term follow-up study needs to be considered to assess the effectiveness and safety of UVA to animals from both physiological and behavioral perspectives. Finally, the suitable application scheme in varied practical situations assumes that UVA can be further popularized and applied. For example, there is a great variability among animals of different ages in terms of the severity of the pathology and the characteristics of the pathogenicity caused by bacteria with pathogenic potential. The present study was conducted using calves aged from 2 to 4 months. Younger calves may be more sensitive to pathogenic microorganisms. These need to be considered and further discussed in future work.

## Figures and Tables

**Figure 1 microorganisms-12-01472-f001:**
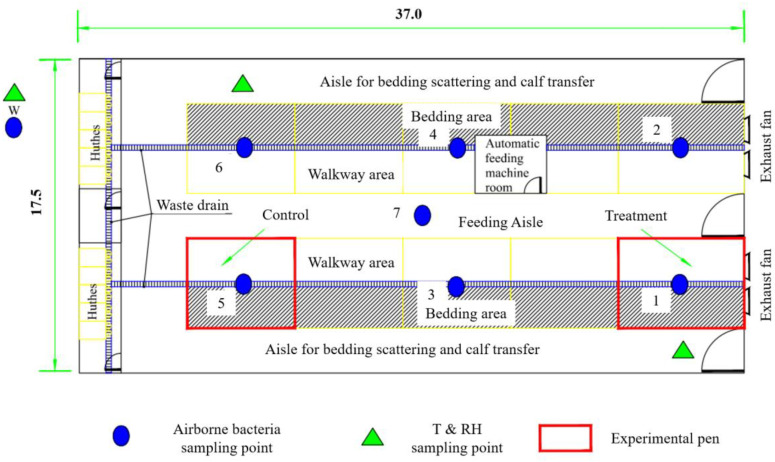
Layout of the experimental calf house and the sampling points (dimension in m).

**Figure 2 microorganisms-12-01472-f002:**
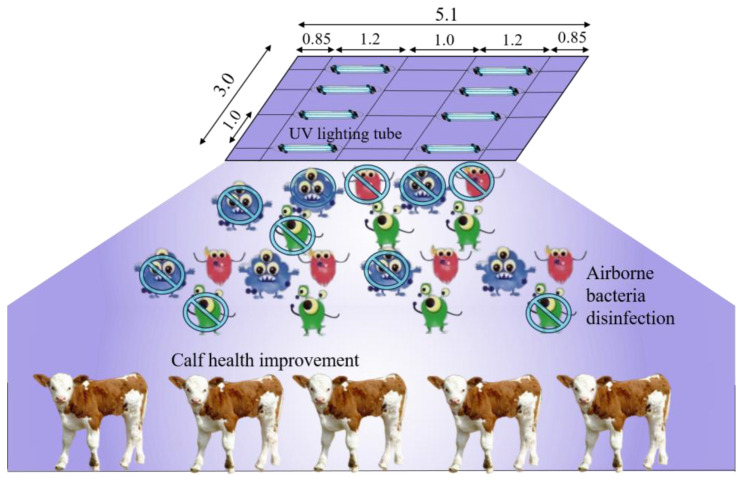
The illustration of UV lighting treatment (dimension in m).

**Figure 3 microorganisms-12-01472-f003:**
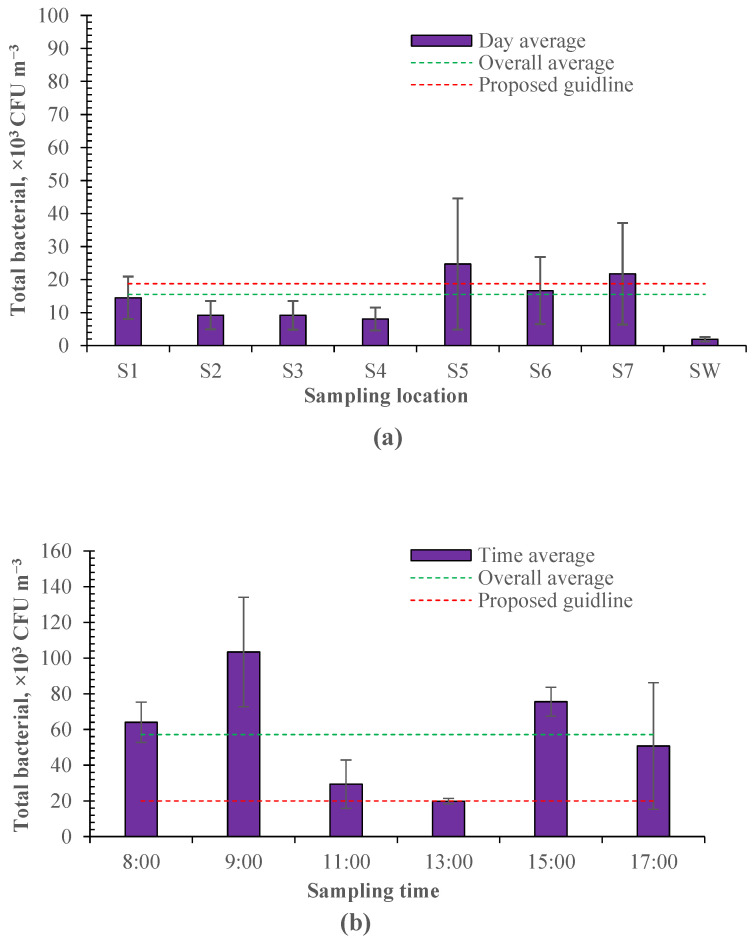
The spatial and temporal variation in airborne bacteria measured in the calf house (*n* = 20 for each error bar). (**a**) Spatial variation. (**b**) Temporal variation.

**Figure 4 microorganisms-12-01472-f004:**
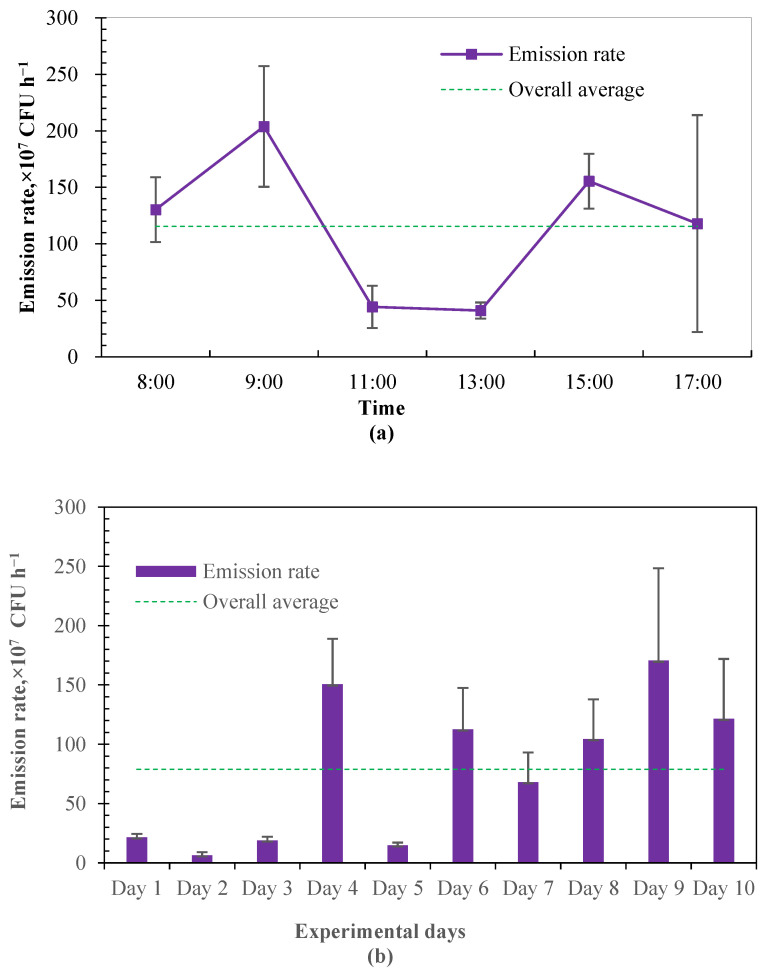
The emission rate of airborne bacteria in the examined calf house (*n* = 10 for each error bar). (**a**) Diurnal variation of emission rate. (**b**) Daily emission rate during experimental period.

**Figure 5 microorganisms-12-01472-f005:**
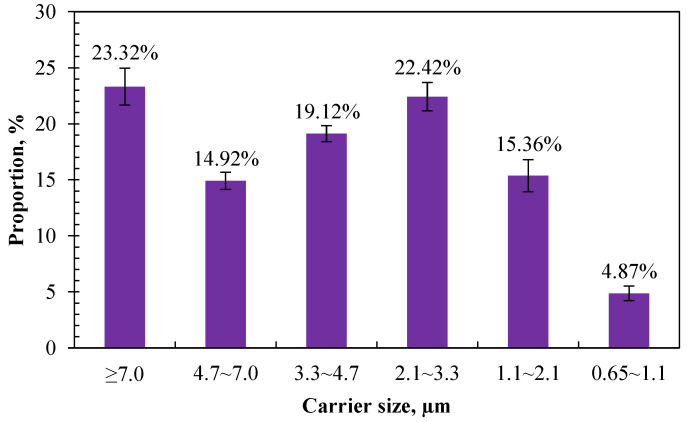
Size distribution of airborne bacteria in calf house (*n* = 20 for each error bar).

**Figure 6 microorganisms-12-01472-f006:**
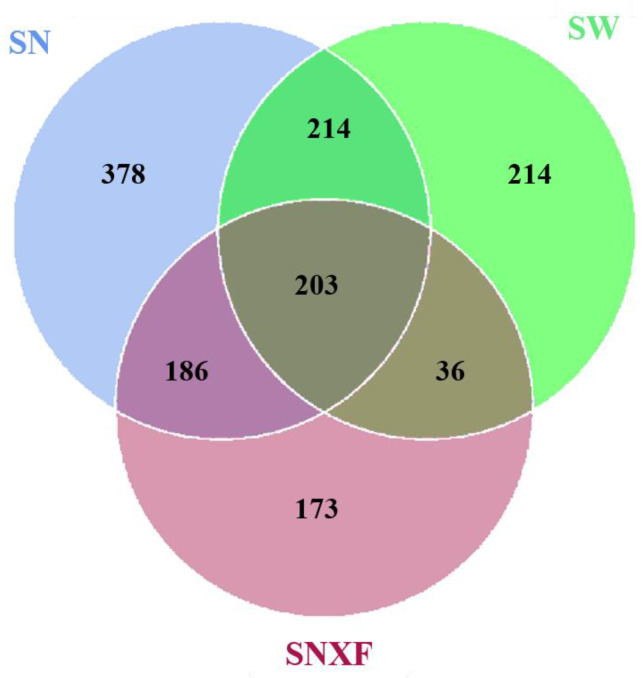
Venn diagram of OUT number for bacterial flora in different samples (SN: indoor air; SW: outdoor air; SNXF: fresh manure).

**Figure 7 microorganisms-12-01472-f007:**
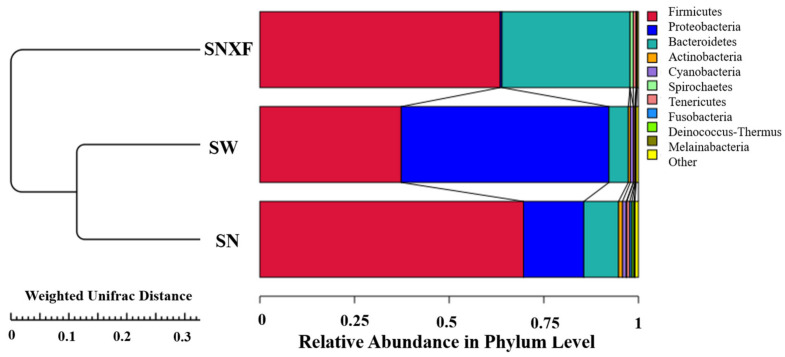
UPGMA clustering tree based on Weighted Unifrac distance (SN: indoor air; SW: outdoor air; SNXF: fresh manure).

**Figure 8 microorganisms-12-01472-f008:**
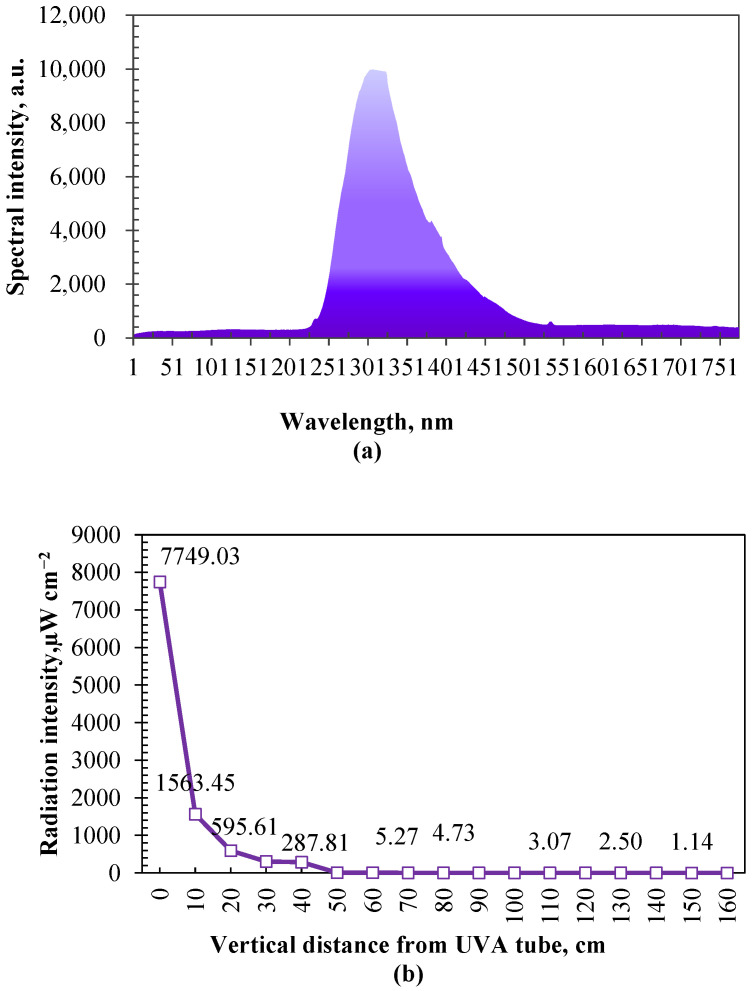
The spectral distribution of UVA tubes used in this study (**a**) and the attenuation cure of radiation intensity with the vertical distance in UVA tube irradiation area (**b**).

**Figure 9 microorganisms-12-01472-f009:**
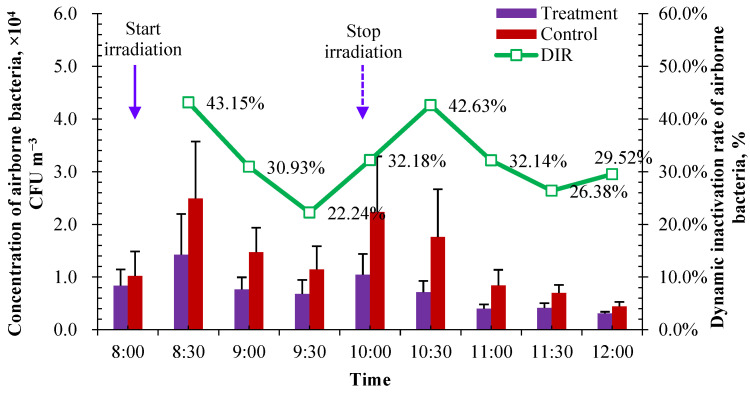
Changes in the concentration of airborne bacteria and DIR during the UVA bactericidal test (*n* = 14 for each error bar).

**Figure 10 microorganisms-12-01472-f010:**
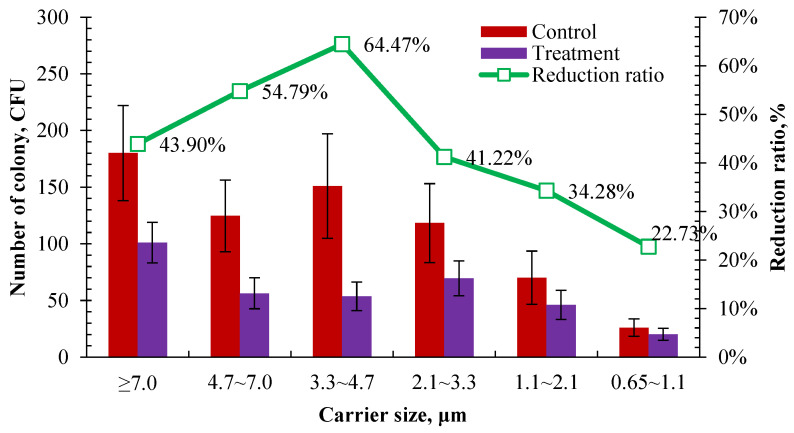
Comparison of the number of colonies for cultured airborne bacteria retained on different stages of the Anderson sampler after 2 h UVA irradiation (*n* = 14 for each error bar).

**Table 1 microorganisms-12-01472-t001:** The description of record behaviors of calves.

No.	Behavior	Description	Reference
1	Grooming	Touching hair with the tongue or mouth on calf’s own body or a neighboring animal.	Adapted from [34]
2	Tongue rolling	Tongue is held in a full or partial circular position or moves in a full or partial circular motion; this can occur when the tongue is held within the border of the lips inside the mouth or extended outside the border of the lips. This cannot occur while any other behaviors are being performed (the tongue is not touching any feed or non-nutritive items), and does not need to be repeated.	[34]
3	Non-nutritive oral manipulation (NNOM)	Licking, chewing, or sucking directed toward a non-nutritive item (includes bars, hutch, bedding, empty bottle), excluding the calf’s own body or that of a neighboring animal. Tongue or lips must be touching a non-nutritive item, or such item must be held inside the mouth.	[34]
4	Itching	The calf rhythmically rubs a part of its body repeatedly against an object.	
5	Cough	A reflex action of throat that produces a sound accompanied by vocal cord vibration.	
6	Cross-sucking (mouth, ear, neck, under belly or body)	The calf is sucking on the mouth, ear, neck, under belly or body of another calf; the sucking movements are performed with the body part in the mouth.	[35]
7	Headbang	The head quickly turns from one direction to another without the body moving, and repeats more than two consecutive times.	

**Table 2 microorganisms-12-01472-t002:** The relative abundance of microbial species in phylum level.

Phyla	Indoor Air	Outdoor Air	Calf Manure
*Firmicutes*	69.65%	37.34%	63.49%
*Proteobacteria*	15.91%	54.87%	0.47%
*Bacteroidetes*	9.19%	5.11%	33.83%
*Actinobacteria*	1.04%	0.54%	0.00%
*Cyanobacteria*	0.99%	0.77%	0.00%
*Tenericutes*	0.75%	0.38%	0.74%
*Fusobacteria*	0.57%	0.08%	0.05%
*Deinococcus-Thermus*	0.54%	0.11%	0.00%
*Melainabacteria*	0.27%	0.18%	0.27%
*Spirochaetes*	0.15%	0.05%	0.83%
Others	0.92%	0.59%	0.33%

**Table 3 microorganisms-12-01472-t003:** The occurrence of behaviors of calves.

NO.	Behavior	Before		After	
		Treatment (%)	Control (%)	Treatment (%)	Control (%)
1	Grooming	11.82 ± 10.28 ^ns^	12.24 ± 9.44	6.12 ± 5.20 ^ns^	8.16 ± 6.46
2	Tongue rolling	22.45 ± 7.64 *	4.08 ± 6.97	26.53 ± 5.00 *	4.08 ± 6.45
3	NNOM	24.49 ± 10.80 ^ns^	30.61 ± 17.36	18.37 ± 14.72 ^ns^	22.45 ± 16.83
4	Headbang	6.12 ± 7.64 *	0	0 *	4.08 ± 10.00
5	Cough	34.69 ± 17.18 ^ns^	32.65 ± 24.35	24.49 ± 10.00 ^ns^*	38.78 ± 16.59
6	Itching	18.37 ± 6.97 ^ns^	20.41 ± 13.94	24.49 ± 16.58 ^ns^*	10.20 ± 10.00

Note: “ns” represents no significant difference between control and treatment (*p* > 0.05); “ns*” represents there is a significant difference in the trend between control and treatment (0.1 > *p* > 0.05); “*” represents there is a significant difference between control and treatment (*p* ≤ 0.05).

## Data Availability

All data are presented in this article in the form of figures and tables.

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
