# Peer review of "A Clean and Health-Care-Focused Way to Reduce Indoor Airborne Bacteria in Calf House with Long-Wave Ultraviolet"

_microorganisms, 2024, doi:10.3390/microorganisms12071472_

Round 1

Reviewer 1 Report

Comments and Suggestions for Authors

Concerning the manuscript by Luyu et al (ref: microorganisms-3058939):

The results are promising and the manuscript has potential to be published.

1.    The title of the paper needs to be rewritten as in its current form it does not reflect the novelty of the manuscript.

2.    Graphical abstract: Due to the novelty of the manuscript, consider adding a graphical abstract. Please, refer to the Instructions for Authors for the journal Microorganisms.

3. Introduction section: If the authors wish, they may cite literature from 2022 on the subject of airborne bacteria. The following paper may be useful: https://doi.org/10.1016/B978-0-323-85206-7.00014-9

4.    Introduction section: Please, avoid listing the objectives.

5.    Materials and methods section: The resolution of Figures 1 and 2 needs to be improved.

6.    Result and discussion section. The resolution of Figures 3, 4, 6, 7 and 8 needs to be improved.

7.    Result and discussion section. Can UVA radiation affect the normal and beneficial microbiota of animals? Discuss and provide a brief clarification

Author Response

Thanks for the comments helping to improve this manuscript. Revisions were made accordingly:

  1. The title was rewritten to reflect the novelty of the manuscript. The new title is “A Clean and Health-care Way to Reduce Indoor Airborne Bacteria in Calf House with the Long-wave Ultraviolet”.
  2. Graphical abstract is added at the end of the manuscript.
  3. Thanks for the information. The literature from 2022 on the subject of airborne bacteria was cited in the introduction section.
  4. these sentences are rewritten to avoid listing the objectives.

5&6. Changes were made to these figures to improve their resolution.

  1. Beneficial microbes on the surface of the skin including epidermis staphylococcus, ‌ staphylococcus capitis, coagulase negative staphylococcus‌, acinetobacter genera, mycobacterium saprophyticus, ‌ hemolytic streptococcus, etc.‌ Very limited information can be found from literature about the effects of UVA radiation on the normal and beneficial microbiota of animals. A section of Future work was added and this is discussed in the future work section

Reviewer 2 Report

Comments and Suggestions for Authors

I attach a file

Author Response

Thank you very much for the valuable comments on improving our manuscript.

1) Microbial sources were added and described in the “Experimental calf house” section. It is solid concrete floor in the calf house and the bedding aera was laid with rice straw. Thus, microbial sources of airborne bacteria mainly include animal itself, excreta of animals, outdoor air bring into the house through ventilation, and indoor facilities in the experimental house such as feed, bedding material, etc. Each pen keeps 7 to 8 calves ranging in age from 2 to 4 months. And their behaviors were observed and analyzed in the section 2.2.3.

2) The authors totally agree with this comment. Thus, outdoor and indoor concentration of airborne bacteria, as well as the ventilation rate were used to assess the emission rate. The peak of emission rate consistent with daily operation of worker such as manure cleaning, feeding. This was discussed in Line 299-305.

3) The six-stage Anderson samplers with disposable nutrient agar plates (every 1000 ml of nutrient agar contains: peptone 10.0 g, beef extract 3.0–5.0 g, sodium chloride 5.0 g, agar powder 12.0–14.0 g) were used to sample airborne bacteria. And it was incubated at 37 °C, for 24 hours. The samples for 16S DNA sequencing were collect by CoriolisÒ µ sampler (Bertin Technologies, France; medium: 15 ml ultrapure water; duration: 10 min; flow rate: 300 L min-1). This is a cyclone liquid sampler widely adopted to collect and analyze bioaerosols in the pandemic of COVID-19. The total genomic DNA from the samples was extracted using the CTAB/SDS method, and the DNA's concentration and purity were assessed by DNA agarose gel electrophoresis. Hereafter, a series of procedures were conducted on the DNA samples, including 16S full-length amplification, the construction of the SMRT Bell library, and sequencing on the PacBio platform.  Detailed please see section 2.2.

4) In this manuscript, the concentration was expressed in CFU/m3, while the emission rate/flux of airborne bacteria was expressed in CFU/h. Fig. 4a plots the emission rate at different time of the day. Number of samples were added in the caption of figures. We change the italics to normal for the bacterial Phylum name. And thank you for the suggestion, analysis of the culturable bacteria at genus and species will be considered in the future work.

Reviewer 3 Report

Comments and Suggestions for Authors

The study aimed to determine the concentration and size of bacteria, their composition, and diversity in a house with calves and to examine the effect of UVA radiation on the concentration and size of bacteria.

The research presented in the manuscript is essential from the point of view of reducing the risk of infections caused in animals and humans by bacteria present in barn rooms.

The manuscript is well-written, and the research methodology is detailed.

However, some issues need the author's attention.

Why were microbial composition and diversity tests not performed after using UVA radiation but only before?

What is the reason for the clear difference in the degree of bacteria removal depending on their size? Why was the most significant reduction observed for bacteria 3.3 – 4.7 microns in size?

Considering that itching spontaneously decreases over time in the control sample, just like NNOM, should such parameters be regarded as the effect of UVA radiation? Isn't there a risk that the observed effects are accidental and not due to UVA radiation?

Author Response

Thank you very much for the valuable suggestions on improving this manuscript.

As the suggested from the reviewer, it is necessary to perform microbial composition and diversity tests after using UVA radiation. While samples didn’t collect for 16s analysis, this is the deficiency of the present study. As this study was conducted in the commercial farm through field measurements to see the potential. This can only be hoped for further follow-up studies in pilot trial. As to the concern on the accidental of itching, this may also require further long-term research. The types of microorganisms attached to aerosols of different particle sizes are indeed different.‌ The sensitivity of different kinds of microorganisms varies because some of them have active repair mechanisms and grow. This may be the reason for the different reduction rate of bacteria on different carrier size.

A section of future work was added to discuss the deficiency of the present study and future work needs to be considered.

Reviewer 4 Report

Comments and Suggestions for Authors

“Each pen keeps 7 to 8 calves, ranging in age from 2 to 4 months”.

The work indicates the age of the calves, however, regarding the severity of the pathology and the characteristics of the pathogenicity caused by bacteria with pathogenic potential, great variability must be assumed. An element that should be considered in the project refers to establishing the criteria that allow measuring the health status of the calves in advance of being exposed to different bacteria. This idea, in my opinion, should be in the discussion of the work

Author Response

Thank you very much for the valuable suggestions on improving this manuscript.

Calves are generally weaned at 6 months of age. Calf death generally occurs before weaning and the main cause are diarrhea and respiratory disease. The highest incidence period is before 2 months of age when poor indoor air quality coupling with the cold stress. The calves from 2 to 4 months are used in the present study. We assumed the same severity of the experimental calves to the pathology and the characteristics of the pathogenicity caused by bacteria with pathogenic potential. The different severity of the pathology on different ages can be considered in the future work. A section of future work was added to discuss the deficiency of the present study and future work needs to be considered.

Round 2

Reviewer 2 Report

Comments and Suggestions for Authors

It is possible that authors change the term desinfection by bectericidal effect, because they didnt test bacterial virulence, and use potential pathogen, because they can talk that all bacteria at genus and species are pathogens, probably some of them.

I would like that review the manuscript and check that genus and species write in italics ( pags. 369-377). Finally why not mention the resuspention as important mechanisms of airborne bacteria??

Author Response

Thanks for the comments helping to improve this manuscript. The term “disinfection” is changed to “bactericidal effect” or “sterilization” according to the context. The manuscript is checked, genus and species are written in italics. The influence of resuspention was added and discussed as below:

“Resuspention is an important mechanism for the transmission of airborne bacteria. As the particle matter would be the carrier of microorganisms, air motion in-out, worker activities and any factor that may cause animals to be more active would increase the amounts of solid and liquid particles, and in turn increase the amounts of airborne bacteria [1, 39].”